# Yamogenin-Induced Cell Cycle Arrest, Oxidative Stress, and Apoptosis in Human Ovarian Cancer Cell Line

**DOI:** 10.3390/molecules27238181

**Published:** 2022-11-24

**Authors:** Justyna Stefanowicz-Hajduk, Anna Hering, Magdalena Gucwa, Monika Czerwińska, J. Renata Ochocka

**Affiliations:** Department of Biology and Pharmaceutical Botany, Medical University of Gdańsk, 80-416 Gdańsk, Poland

**Keywords:** steroidal saponins, SKOV-3 line, in vitro, caspases, mitochondrial potential, tumor necrosis factor superfamily members, intrinsic and extrinsic apoptotic pathway

## Abstract

Steroidal saponins are a group of compounds with complex structures and biological activities. They have anti-inflammatory, antimicrobial, fungicidal, and antitumor properties. Yamogenin is one of the spirostane saponins and occurs in *Trigonella foenum-graecum*, *Asparagus officinalis*, and *Dioscorea collettii*. It is a stereoisomer of diosgenin—a well-known compound whose activity and mechanisms of action in cancer cells are determined. However, the antitumor effect of yamogenin is still little known, and the mechanism of action has not been determined. In this study, we evaluated the effect of yamogenin on human ovarian cancer SKOV-3 cells in vitro by determining the cellular factors that trigger cell death. The viability of the cells was assessed with a Real-Time xCELLigence system and the cell cycle arrest with flow cytometry. The activity of initiator and executioner caspases (-8, -9, and -3/7) was estimated with luminometry and flow cytometry, respectively. The mitochondrial membrane depolarization, the level of oxidative stress, and DNA damage in the yamogenin-treated cells were also evaluated by flow cytometry. Genes expression analysis at the mRNA level was conducted with Real-Time PCR. Bid activation and chromatin condensation were estimated with fluorescent microscopy. The obtained results indicate that yamogenin has cytotoxic activity in SKOV-3 cells with an IC_50_ value of 23.90 ± 1.48 µg/mL and strongly inhibits the cell cycle in the sub-G1 phase. The compound also triggers cell death with a significant decrease in mitochondrial membrane potential, an increase in the level of oxidative stress (over two times higher in comparison to the control), and activation of caspase-8, -9, -3/7, as well as Bid. The results of genes expression indicate that the Tumor Necrosis Factor (TNF) Receptor Superfamily Members (TNF, TNFRSF10, TNFRSF10B, TNFRSF1B, and TNFRSF25), Fas Associated via Death Domain (FADD), and Death Effector Domain Containing 2 (DEDD2) were significantly upregulated and their relative expression was at least two times higher than in the control. Our work shows that yamogenin induces apoptosis in ovarian cancer cells, and both the extrinsic and mitochondrial—intrinsic pathways are involved in this process.

## 1. Introduction

Steroidal saponins are widely distributed in plant species. They are high-molecular-weight compounds that consist of aglycone and sugar moiety. They can have a 5-ring furostane or a 6-ring spirostane skeleton connected with D-glucose, D-galactose, L-rhamnose, D-xylose, L-arabinose, or D-glucuronic acid [1]. The distribution of these metabolites is mainly in Monocotyledones plants, among which Liliaceae, Agavaceae, and Dioscoreaceae are the main source of these steroids [2]. The compounds have many pharmacological and biological properties. One of these is antimicrobial, anti-inflammatory, antitumor, fungicidal, insecticidal, antifeedant, and molluscicidal activity [2,3,4,5,6,7,8,9,10]. Furthermore, many of them are precursors used by the pharmaceutical industry in the synthesis of steroid hormones—progesterone and cortisone derivatives [2]. Generally, saponins are a structurally diverse group, and they are characterized by unique biological and physicochemical properties—hemolytic, foaming, and detergent abilities [11]. The compounds form complexes with cell membrane cholesterol, creating pores and changes in carbohydrate portions on the cell surface [1].

Recently, steroidal saponins have been extensively studied due to their significant cytotoxic activities on cancer cells. Among them are aglycones and their derivatives, such as diosgenin, dioscin, gracillin, sarsasapogenin, pennogenin, yamogenin, aginoside, asparagoside, prosapogenin, icogenin, gitonin, and polyphyllin D [1]. These compounds were isolated from plant species and tested on different kinds of cancer cells—murine lymphocytic leukemia P-388 cell line, human epidermoid carcinoma KB, murine lymphoma L-1210, human acute promyelocytic leukemia HL-60, human lung carcinoma A549, human cervical cancer HeLa, human hepatocellular carcinoma HepG2 cells, and many other [2,12,13,14,15,16,17,18]. Up to now, the cytotoxic activity of diosgenin and its glycoside dioscin is very well known, and their mechanisms of action have been determined in many cancer cells [19,20,21,22]. Diosgenin stereoisomer is yamogenin ((25S)-spirost-5-en-3beta-ol), which is also called neodiosgenin (Figure 1) and occurs in *Trigonella foenum-graecum* [23], *Solanum violaceum* [24], *Dioscorea collettii* [25], and *Asparagus officinalis* [26]. Lu et al. tested yamogenin glycoside on a few cancer cell lines—human breast cancer MCF-7, lung carcinoma A549, hepatocellular carcinoma HepG2, and gastric adenocarcinoma MGC-803, and obtained IC_50_ (inhibitory concentration) values were between 20 and 30 µg/mL [27]. Moreover, yamogenin can inhibit triacylglyceride accumulation and suppress mRNA expression of fatty acid synthesis-related genes in HepG2 [28]. Our previously reported study showed that yamogenin has cytotoxic activity on human cervical HeLa and ovarian SKOV-3 cells in vitro [29]; however, its mechanism of action, according to our knowledge, has not been determined. As yamogenin and diosgenin have similar chemical structures, as shown in Figure 1, it is highly probable that their biological activities and anticancer pathways can be comparable. In addition, both compounds are present in the same popular plant materials used according to their nutritional and pro-healthy properties [23,24,25]. It should be emphasized that yamogenin is a compound that is relatively poorly analyzed both in terms of pharmacology and anticancer activity in vitro. The significance of clarifying the effect of yamogenin in tumor cells may be the basis for further study in these fields. 

In this study, we present the activity of yamogenin on human cancer ovarian SKOV-3 cells in vitro. The mechanism of action leading to cell death induced by the compound was determined. We postulate that yamogenin induces cell death via apoptosis, and both the intrinsic and extrinsic pathways are involved in this process. 

## 2. Results

### 2.1. Yamogenin Decreases the Viability of the Ovarian Cancer SKOV-3 Cells

To estimate the proliferation and viability of ovarian cancer SKOV-3 cells as well as non-cancer keratinocytes treated with yamogenin, we performed experiments with Real-Time Cell Analyzer (xCELLigence system). The system enables evaluation of the cellular viability based on changes in electric impedance, which can be observed in real-time and continuously. The obtained results showed that yamogenin decreased the viability of the cancer SKOV-3 cells as well as non-cancer HaCaT. The IC_50_ values calculated by the system were 23.90 ± 1.48 and 16.40 ± 1.41 µg/mL, respectively (Figure 2A and Figure 3A). Furthermore, we show the changes in IC_50_ values of yamogenin every 30 min during the whole time of conducted experiments (24 h) (Figure 2B and Figure 3B), as well as the level of changes in the cytotoxic effect (slope) of yamogenin for every used compound concentration (Figure 2C and Figure 3C). In the case of SKOV-3 cells, the increase in the cytotoxic effect of yamogenin was visible above 10 µg/mL of the compound concentration, while in HaCaT cells, this effect occurred above 15 µg/mL. 

To confirm the yamogenin effect on the viability and morphology of SKOV-3 cells, we performed fluorescent staining using Hoechst 33342 dye. The microscopic observation before staining revealed an increase in the number of cells with loss of cell volume and detached from the plate surface (Figure 4C,D). After staining, we also observed significant fragmentation and condensation of chromatin in nuclei of the cells treated with yamogenin in comparison with the control (Figure 4G,H,K,L).

### 2.2. Yamogenin Induces Cell Cycle Arrest in Sub-G1 Phase of SKOV-3 Cells

The cell cycle of SKOV-3 treated with yamogenin was assessed with flow cytometry. The results showed that the compound induced cell cycle arrest in the sub-G1 phase. The results were 12.6 ± 0.34, 11.88 ± 0.21, 12.05 ± 0.58, 21.8 ± 1.37, and 28.9 ± 2.51% for the control (ethanol 0.7%), and yamogenin concentrations of 10, 20, 50, and 70 µg/mL, respectively (Figure 5).

### 2.3. Yamogenin Depolarizes Mitochondrial Membrane in SKOV-3 Cells

To estimate the changes in the polarization of the mitochondrial membrane in SKOV-3 cells treated with yamogenin, we used flow cytometry. The obtained results showed that yamogenin caused a decrease in MMP (mitochondrial membrane potential), and this effect was very significant at concentrations of the compound above 20 µg/mL. The percentage of live/depolarized cells was 4.1 ± 0.75, 5.99 ± 0.96, 18.48 ± 3.16, 46.28 ± 1.44, and 67.70 ± 2.67 for the control, and yamogenin concentrations 10, 20, 50, and 70 µg/mL, respectively (Figure 6).

### 2.4. Yamogenin Induces Oxidative Stress in SKOV-3 Cells

The cells were treated with different concentrations of yamogenin, and the flow cytometry analysis revealed that the compound increased the level of cellular oxidative stress and its relative value was over two times higher than in the control cells. The results were 2.57 ± 0.21 and 2.70 ± 0.24 for concentrations of 50 and 70 µg/mL, respectively, while for the cells treated with ethanol (0.7% (*v*/*v*), a control), this level was 1.0 ± 0.08 (Figure 7).

### 2.5. Yamogenin Triggers Up-Regulation of the Tumor Necrosis Factor Receptor Superfamily (TNFRSF) Members Genes 

To estimate the up- and down-regulation of selected genes, we prepared an RT-PCR analysis. The obtained results showed that yamogenin at a concentration of 40 µg/mL significantly increased expression at mRNA level (over two times higher than control) of 35 genes related to cell death and down-regulated three genes. The most changes were observed in the case of BCL2 Associated Agonist of Cell Death (BAD), BCL10 Immune Signaling Adaptor (BCL10), BCL2 Related Protein A1 (BCL2A1), BCL2 Like 13 (BCL2L13), BCL2 Interacting Killer (BIK), X-Linked Inhibitor of Apoptosis (XIAP), BCL2 Interacting Protein 3 (BNIP3), BCL2 Interacting Protein 3 Like (BNIP3L), BCL2 Family Apoptosis Regulator BOK (BOK), Death Effector Domain Containing 2 (DEDD2), Fas Associated via Death Domain (FADD), Leucine Repeat Death Domain (LRDD), Nuclear Factor Kappa B Subunit 2 (NFKB2), NFKB Inhibitor Alpha (NFKBIA), NFKB Inhibitor Zeta (NFKBIZ), Phorbol-12-Myristate-13-Acetate-Induced Protein 1 (PMAIP1), Tumor Necrosis Factor (TNF), and TNF Receptor Superfamily Member 25 (TNFRSF25) gene. The up-regulation was over 2.5 times higher than the control (Figure 8), while the down-regulation was lower than 0.5 for Fas Cell Surface Death Receptor (FAS) (0.48 ± 0.002), TNF Receptor Superfamily Member 21 (TNFRSF21) (0.49 ± 0.003), and TNF Superfamily Member 10 (TNFSF10) (0.25 ± 0.001). 

### 2.6. Yamogenin Increases the Activity Level of Caspases-3/7/8/9 in SKOV-3 Cells

To estimate the level of activity of initiator and executioner caspases, we used luminometry and flow cytometry, respectively. In luminometric analysis, we observed a significant increase in caspase-8 and -9 activities. The results revealed that the relative activity level of caspase-8 was 1.14 ± 0.18, 1.71 ± 0.05, 3.43 ± 0.09, and 3.45 ± 0.09 times higher than the control at yamogenin concentrations of 10, 20, 50, and 70 µg/mL, respectively. The relative activity level of caspase-9 was also much higher than the control and was 1.12 ± 0.14, 3.08 ± 0.34, 5.09 ± 0.47, and 5.49 ± 0.36 at yamogenin concentrations of 10, 20, 50, and 70 µg/mL, respectively (Figure 9). 

The obtained results from flow cytometry analysis indicated that the activity level of caspase-3/7 increased in the cells treated with yamogenin above concentration of 20 µg/mL. When caspase-3/7 is active, the released fluorescent dye (previously linked to a DEVD peptide substrate) labels the DNA of the apoptotic cells. Thus, the method enables estimation of the amount of apoptotic and dead cells in the treated population. The percentage of late apoptotic cells in this experiment was 2.17 ± 0.13, 2.90 ± 0.42, 18.68 ± 0.67, 50.08 ± 1.99, and 55.95 ± 2.11 for the control, and 10, 20, 50, and 70 µg/mL of yamogenin, respectively. The number of dead cells was 1.88 ± 0.29, 1.58 ± 0.37, 5.61 ± 0.25, 16.83 ± 1.37, 18.75 ± 1.98 % for the control, and 10, 20, 50, and 70 µg/mL of yamogenin, respectively (Figure 10).

### 2.7. Yamogenin Activates Bid in SKOV-3 Cells

The cells were incubated with yamogenin at concentrations of 20 and 40 µg/mL. After 24 h, the cells were treated with primary anti-Bid antibody and secondary goat anti-rabbit IgG Alexa Fluor 594. The cellular nuclei were stained with Hoechst 33342 dye and visualized with fluorescent microscopy. The obtained results showed that the intensity of red fluorescence is much stronger in the yamogenin-treated cells than in the control, which indicates that Bid was activated in SKOV-3 cells (Figure 11).

### 2.8. Yamogenin Induces H2A.X Activation in SKOV-3 Cells

The cells were treated with yamogenin or etoposide as a positive control to assess the amount of the cells with activated H2A.X. Active form of a histone H2A.X (phospho histone H2A.X, known as γH2A.X)—an indicator of DNA damage is generally detected with conjugated antibodies—a phospho-specific anti-phospho-histone H2A.X (Ser139)-Alexa Fluor555 and an anti-histone H2A.X-PECy5. The total level of histone H2A.X was measured in SKOV-3 cells treated with yamogenin. The significant changes in the percentage of the cells with phospho H2A.X we observed at yamogenin concentrations of 50 and 70 µg/mL in comparison to the control, and this amount was 13.54 ± 2.61 and 20.2 ± 2.08%, respectively. In the case of etoposide, the percentage of the cells with phospho H2A.X was 46.2 ± 3.89 (Figure 12).

## 3. Discussion

In our study, we examined the effect of yamogenin on human ovarian cancer SKOV-3 cells. The tested compound showed cytotoxic activity on the cell line, and this effect was dose- and time-dependent. Furthermore, the compound caused cell death. Another experiment revealed that yamogenin suppressed cell proliferation and induced strong inhibition of the cell cycle in the sub-G1 phase. The arresting of the cell cycle in this phase may indicate that apoptosis is the main way of cellular death. To confirm this hypothesis, we performed experiments to show changes in the activity level of initiator and executioner caspases. In this case, caspase-8, -9, and -3/7 were upregulated, and these changes were significant in comparison to the control. These effects were dose-dependent.

Apoptosis is one of the main types of regulated cell death and is characterized by morphological and biochemical changes in cells. Some of the most well-known hallmarks of apoptosis are cell shrinkage, plasma membrane blebbing, condensation of chromatin, and DNA fragmentation [30]. All these changes were observed in our experiment after treating the cells with different concentrations of yamogenin. After staining the cells with fluorescent Hoechst dye, a clear formation of condensed nuclei with the fragmentation of DNA was observed. 

The apoptotic process can take place through the activation of external and/or internal stimuli [31]. The external way starts from the interaction of cell death receptors (DR) with external factors such as drugs, UV, radiation, and pathogens infections. This pathway involves transmembrane receptors that are members of the tumor necrosis factor (TNF) receptor superfamily, characterized by cysteine-rich extracellular domains, as well as cytoplasmic death domains [30,32,33]. The well-known receptors and ligands include FasL/FasR, TNF-α/TNFR1, Apo3L/DR3, Apo2L/DR4, and Apo2L/DR5 [30,34,35]. The first two models—FasL/FasR, TNF-α/TNFR1 are best described. In this model, the receptors can be activated by the Fas ligand (FasL) and tumor necrosis factor-related apoptosis-inducing ligand (TRAIL). The apoptotic signal is induced when these ligands bind to DR and TRAIL receptors. The binding of FasL to FasR triggers the binding of the adapter protein FADD as well as the binding of the TNF ligand to the TNF receptor results in the binding of the adapter protein TNF-associated death domain (TRADD) with the recruitment of FADD and receptor-interacting protein (RIP) [36,37]. In the next step, the death-inducing signaling complex (DISC) is formed, and the activation of procaspase-8 is observed [38]. Furthermore, DEDD2 encodes a protein that is associated with the extrinsic pathway of apoptosis and may target caspase-8 and -10 to the nucleus and regulate nuclear events—degradation of intermediate filaments during the cell death [39]. In our work, Real-Time PCR analysis revealed that the expression of TNFRSF members genes in yamogenin-treated SKOV-3 cells was significantly upregulated, as well as mRNA expression of FADD and DEDD2, which was over two times higher in comparison to the control. Simultaneously, we did not observe the increased expression of FAS, which may indicate that yamogenin preferentially induces apoptosis by one of the groups of death receptors. Furthermore, the activity level of caspase-8 measured with luminometry was over three times higher than in the untreated cells, which confirms that the extrinsic way of apoptosis plays an important role in the cell death of SKOV-3. 

The extrinsic pathway may enhance the intrinsic way, and the protein that links these two pathways is Bid. Its activated form translocates to the mitochondria and, after interaction with Bcl-2-associated X (Bax) and Bcl-2 antagonist or killer (Bak) proteins, causes mitochondrial outer membrane permeabilization (MOMP) and a decrease in mitochondrial membrane potential (MMP) is observed [31,40,41]. In our work, we show a significant decrease in MMP in SKOV-3 cells treated with yamogenin, as well as activation of Bid, and this effect was the strongest at the highest used concentrations of the compound. 

The intrinsic pathway of apoptosis, called the mitochondrial pathway, occurs as an effect of cellular stress caused by DNA damage or endoplasmic reticulum stress [30]. We observed an increase in oxidative stress level in the yamogenin-treated cancer cells, as well as DNA damage. H2A.X is a member of the histone H2A family, and its phosphorylation at serine 139 is a hallmark of DNA damage [42]. Detection by flow cytometry is possible with conjugated antibodies to measure the total level of histone H2A.X in a tested population of cells. In SKOV-3 cells treated with yamogenin, DNA damage occurred; however, the number of cells with damaged DNA was up to 20%, while in the case of etoposide treatment, the percentage of the cells with phospho-histone H2A.X was almost 50%. Nevertheless, these results indicate that DNA damage is one of the factors taking part in the death of SKOV-3 cells.

The final stage of apoptosis is activation of effector caspase-3 and -7 by initiator caspase-9. The latter caspase cleaves execution caspases through proteolysis, which activates other execution caspases in a feedback system [43]. Our study showed that caspase-9, as well as effector caspase-3/7, were activated. The activity level of caspase-9, measured by luminometry, was three times higher in SKOV-3 cells treated with a yamogenin concentration of 20 µg/mL and five times higher at concentrations of 50 and 70 µg/mL in comparison to the untreated cells. This confirms the participation of the mitochondrial pathway in the apoptotic process of the ovarian cell line. The proposed scheme of the death process in SKOV-3 cells treated with yamogenin is shown in Figure 13.

Saponins are potential anticancer compounds with different mechanisms of action. They can trigger cell death through apoptotic or non-apoptotic stimulation. One of the more well-known steroidal saponins is diosgenin, which inhibited the cell cycle in the G1 phase and activated p53 in osteosarcoma cells [19]. In that study, the expression of caspase-3 mRNA was not modified, but hsp70 mRNA expression was strongly increased. The hsp proteins have the function of regulating cellular homeostasis and promoting survival. They are also associated with p53 [44]. Moreover, diosgenin can trigger similar results in other cell lines. For example, in laryngocarcinoma HEp-2 and melanoma M4Beu cells, diosgenin also strongly inhibited proliferation, blocked the cell cycle as in osteosarcoma cells, and activated p53; however, cell cycle arrest was observed in S and G2/M phases [45]. The compound induced apoptosis by a mitochondrial pathway in both lines (HEp-2 and M4Beu) with a fall of mitochondrial potential, caspase-9 and -3 activation, nuclear localization of apoptosis-inducing factor (AIF), and cleavage of poly (ADP-ribose) polymerase (PARP) [45]. The HeLa cells were treated with diosgenin, and the mitochondrial pathway of apoptosis also occurred, but without the participation of the extrinsic way of the cell death [46]. Next, in human leukemia K562 cells, this steroidal saponin can induce as well G2/M cell cycle arrest and apoptosis with disruption of intracellular Ca^2+^ homeostasis, mitochondrial membrane depolarization, ROS production, and caspases activity [20]. All these works clearly show that diosgenin triggers cell death through intrinsic, mitochondrial way of apoptosis. In turn, another study indicates that different mechanisms of action of diosgenin and other saponins can be observed. In breast cancer cells, estrogen receptor positive and negative, diosgenin affected prosurvival Akt-mediated NF-κB and mitogen-activated protein kinase (MAPK) signaling pathways, caused G1 cell cycle arrest and downregulated cyclin D1, cdk-2, and cdk-4 expression resulting in the inhibition of cell proliferation and induction of apoptosis [21]. Dioscin—a diosgenin glycoside—induced apoptosis in human ovarian cancer SKOV-3 cells in a dose-dependent manner, increased caspase-3 and -9 activity, the protein expression of Bax, and suppressed cell viability by regulating the PI3K/AKT/MAPK signaling pathways [47]. Next, icogenin, isolated from *Dracaena draco*, was tested on myeloid leukemia HL-60 cell line and induced nuclear changes, fragmentation of poly(ADP-ribose) polymerase-1, and led to apoptosis [1,16]. Another steroidal saponin, polyphyllin D from *Paris polyphylla*, was tested on human lung cancer NCI-H460 cells and caused upregulation of endoplasmic reticulum (ER) stress-related proteins, disruption of mitochondrial membrane, and activation of caspase-9 and -3 [15]. Another study revealed that the mixture of steroid glycosides (balanitin-6 and -7) from *Balanites aegyptiaca* showed cytotoxic activity on human lung cancer A549 and glioblastoma U373 cells with a decrease in intracellular ATP and disorganization of the actin cytoskeleton. In this case, cell death was not associated with apoptosis [48]. 

Yamogenin, in comparison to diosgenin and other steroidal saponins, has a similar effect by induction of cell cycle arrest and apoptosis with the significant role of the intrinsic mitochondrial pathway, in which mitochondrial membrane depolarization, ROS production, and caspase-9/3/7 activation are observed. The extrinsic pathway of apoptosis was, in turn, confirmed in the case of pennogenin glycosides obtained from *Paris quadrifolia* [49]. In the study, the compounds triggered apoptosis by death receptors, activated caspase-8 and caspases-3/7, and also induced the intrinsic way with depolarization of the mitochondrial membrane, Bid, and caspase-9 activation [49]. 

Since research on yamogenin's effect on cancer cells is very limited, other cellular factors that can be involved in the death induced by yamogenin, as well as the compound action in cells and organs in vivo, should be determined in the future. 

## 4. Materials and Methods

### 4.1. Preparation of Yamogenin Solution

Yamogenin was obtained from Merck Millipore (Burlington, MA, USA) and dissolved in absolute ethanol at the concentration of 10 mg/mL with the use of an ultrasonic water bath (50 Hz).

### 4.2. Cell Culture

The human ovarian cancer SKOV-3 cell line and human keratinocytes HaCaT were obtained from the American Type Culture Collection (ATCC, Manassas, VA, USA). The SKOV-3 cell line was cultured in McCoy’s Medium, and HaCaT cells were maintained in Dulbecco’s Modified Eagle’s Medium (DMEM). Both media were supplemented with 100 units/mL of penicillin, 100 µg/mL of streptomycin, and 10% (*v*/*v*) fetal bovine serum (FBS) (Merck Millipore). The cells were incubated at 37 °C and 5% CO_2_.

### 4.3. Real-Time Cell Analysis

Real-Time Cell Analyzer (xCELLigence system, Acea Biosciences, San Diego, CA, USA) was used to estimate the effect of yamogenin on SKOV-3 and HaCaT cell lines. This system enables the monitoring of cell viability and proliferation in real-time and continuously. The cells were seeded (2 × 10^4^ cells/well) in E-plates 16 (Acea Biosciences, San Diego, CA, USA) for 24 h, then yamogenin was added to the plate wells at concentrations of 1, 5, 10, 15, 20, 30, 40, and 50 µg/mL, as previously described [29]. During 24 h of incubation of SKOV-3 cells with the compound, the changes in viability, proliferation, and morphology were detected by the xCELLigence system. The values of IC_50_ calculated during the experiment, as well as slope, were obtained with RTCA Software v.1.2.1. (Acea Biosciences, San Diego, CA, USA). The experiments were performed in duplicate, in three independent repeats (n = 6). 

### 4.4. Hoechst 33342 Staining of SKOV-3 Cells Treated with Yamogenin

The SKOV-3 cells were seeded in a 12-well plate with coverslips (1 × 10^5^ cells/well) and incubated with yamogenin at concentrations of 20, 50, and 70.0 µg/mL. The concentration of ethanol in the wells did not exceed 0.7% (*v*/*v*). After 24 h, the cells on the coverslips were washed with PBS buffer and stained with Hoechst 33342 dye (ThermoFisher Scientific, Waltham, MA, USA). The coverslips were observed under the fluorescence microscope (350/461 nm). The experiment was repeated twice.

### 4.5. Cell Cycle Analysis of SKOV-3 Cells Treated with Yamogenin

The SKOV-3 cells were seeded in a 6-well plate (5 × 10^5^ cells/well) and incubated with yamogenin in the concentration range of 10.0–70.0 µg/mL for 48 h. The concentration of ethanol added to the cells did not exceed 0.7% (*v*/*v*). Next, the cells were prepared with Muse Cell Cycle Assay Kit (Merck Millipore), and the number of cells in each phase of the cell cycle was determined by Muse Cell Analyzer (Merck Millipore). The experiment was repeated three times.

### 4.6. Estimation of Mitochondria Membrane Depolarization in SKOV-3 Cells Treated with Yamogenin

The SKOV-3 cells were seeded in a 12-well plate (1 × 10^5^ cells/well) and incubated with yamogenin at concentrations of 10.0–70.0 µg/mL. The concentration of ethanol added to the cells did not exceed 0.7% (*v*/*v*). After 24 h of the treatment, the cells were stained with Muse MitoPotential Assay Kit (Merck Millipore), and determination of the percentage of depolarized/live and dead cells was conducted with Muse Cell Analyzer. All the experiments were independently repeated three times.

### 4.7. Reactive Oxygen Species (ROS) Production in SKOV-3 Cells Treated with Yamogenin

The SKOV-3 cells (1 × 10^5^ cells/well, 12-well plate) were treated with yamogenin in the concentration range of 10.0–70.0 µg/mL. The concentration of ethanol added to the cells did not exceed 0.7% (*v*/*v*). After 24 h of incubation, the cells were stained with Muse Oxidative Stress Kit (Merck Millipore) and analyzed with Muse Cell Analyzer. The experiments were conducted in three independent repeats.

### 4.8. RT-PCR Analysis of Genes Expression in SKOV-3 Cells Treated with Yamogenin

The SKOV-3 cells were incubated with yamogenin at a concentration of 40.0 µg/mL for 24 h. Next, the total RNA of the cells was isolated using the RNeasy Mini Kit (Qiagen, Hilden, Germany), and the concentration of RNA was estimated with Agilent Technologies 4200 TapeStation (Agilent Technologies, Santa Clara, CA, USA), according to the manufacturer’s protocol. The Maxima First Strand cDNA Synthesis Kit (ThermoFisher Scientific, Scientific, Waltham, MA, USA) was used for cDNA synthesis.

cDNA was applied on the TaqMan Array Human Apoptosis Fast 96-well plates (ThermoFisher Scientific). Each plate contains 92 assays for genes associated with cell death and four assays for control genes (Appendix A). The PCR reactions were performed in StepOnePlus Real-Time PCR System (ThermoFisher Scientific). The data were obtained in three independently repeated experiments and analyzed with StepOne software v. 2.3. (ThermoFisher Scientific, Scientific, Waltham, MA, USA).

### 4.9. Caspases-3/7/8/9 Activity in SKOV-3 Cells Treated with Yamogenin

The cells were seeded in 12-well plate (1 × 10^5^ cells/well) and treated with yamogenin at concentrations of 10.0–70.0 µg/mL. The concentration of ethanol added to the cells did not exceed 0.7% (*v*/*v*). After 24 h of treatment, caspase-3/7 activation was measured, and an estimation of the apoptotic status of the cells was conducted. The cells were stained with a fluorescent reagent that contained a DNA-binding dye linked to a DEVD peptide substrate. The dye is released from the complex when caspase-3/7 is active. A cell marker (7-AAD) was also used in the assay as a marker of dead cells. The cells were analyzed with flow cytometry (Muse Cell Analyzer). The experiments were performed in three independent repeats.

The caspases-8 and -9 activity level in the cells was determined with Caspase-Glo 8 or 9 Assay Kit (Promega, Madison, WI, USA) and Glomax Multi + Detection System (Promega). The cells were seeded in 96 well plates (1 × 10^4^ cells/well), and after 24 h of incubation, they were treated with yamogenin at concentrations of 10–70 µg/mL for 24 h. The experiments were performed in three independent repeats.

### 4.10. Assessment of Bid Activation in SKOV-3 Treated with Yamogenin

The cells on the coverslips (in a 6-well plate, 5 × 10^5^ cells/well) were incubated with yamogenin at concentrations of 20 and 40 µg/mL for 24 h. The concentration of ethanol did not exceed 0.4% (*v*/*v*) (a control). Then, the cells were fixed with 4% paraformaldehyde (*v*/*v*) for 10 min and permeabilized with 0.1% Triton in PBS (*v*/*v*) for 30 min. After blocking with 5% BSA (*w/v* in PBS) for 60 min, the cells were incubated overnight at 4 °C with primary rabbit polyclonal IgG anti-Bid antibody (1:300 in PBS with 2% FBS (*v*/*v*) and 1% BSA (*w/v*), Merck Millipore). Then, the cells were washed three times with PBS, and a secondary goat polyclonal anti-rabbit IgG Alexa Fluor 594 antibody (ThermoFisher Scientific) was used (1:1000) and incubated with the cells for 1 h at RT. After this incubation, the cells were washed three times with PBS and stained with Hoechst 33342 dye (3 µg/mL) for 20 min at RT. The cells on the coverslips were observed under a fluorescent microscope with filters D (355–425 nm) and N21 (515–560 nm) (Leica, Wetzlar, Germany). 

### 4.11. Estimation of H2A.X Activation in SKOV-3 Cells Treated with Yamogenin

The SKOV-3 cells were seeded in a 12-well plate (1 × 10^5^ cells/well) and incubated for 24 h in a concentration range of 10.0–70.0 µg/mL. The concentration of ethanol added to the cells did not exceed 0.7% (*v*/*v*). The positive control—etoposide—was used at a concentration of 20 μM. After 24 h, the cells were stained with H2A.X Activation Dual Detection Kit (Merck Millipore) according to the manufacturer’s instruction and analyzed with flow cytometry (Muse Cell Analyzer). The experiment was repeated three times.

### 4.12. Statistical Analysis

Statistical data were obtained using the STATISTICA 12.0 software package (StatSoft. Inc., Tulsa, OK, USA). All data were expressed as mean values ± standard deviation (SD). The Student’s *t*-test was used to compare the results with the control sample. The statistical significance was set at *p* < 0.05. 

## 5. Conclusions

In our work, we demonstrate the antitumor effect of yamogenin in human ovarian cancer SKOV-3 cells in vitro. The compound triggers cell cycle arrest, overproduction of oxidative stress, mitochondrial membrane depolarization, caspase-8, -9, -3/7 and Bid activation, and DNA damage. The cell death is activated by TNF receptor superfamily members and is mediated by both the extrinsic and intrinsic pathways of apoptosis. 

## Figures and Tables

**Figure 1 molecules-27-08181-f001:**
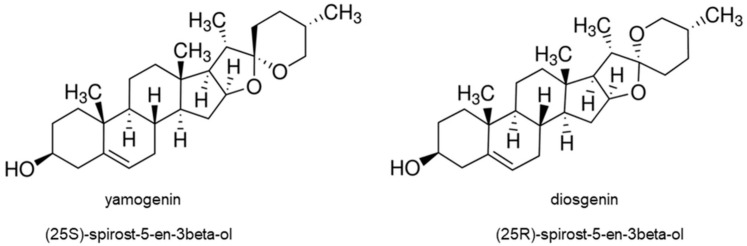
The structure of yamogenin and diosgenin.

**Figure 2 molecules-27-08181-f002:**
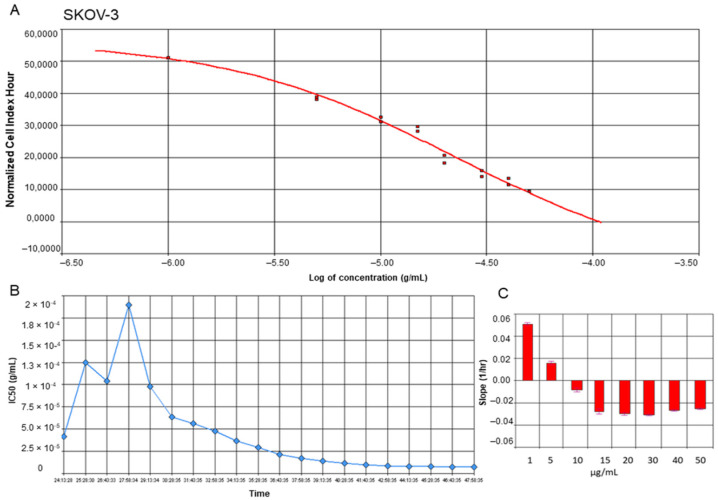
RTCA profiles of cell index (**A**), IC_50_ values (**B**), and slope of changes (**C**) in cytotoxic effect of yamogenin on the SKOV-3 cells. The cells were incubated with the compound for 24 h, and normalized cell index vs. concentration of yamogenin was calculated by RTCA system (**A**). The IC_50_ values of yamogenin were obtained at every time point of the experiment (every 30 min) (**B**). The slope of changes in cytotoxic effect of yamogenin was calculated for every used concentration of the compound during the experiment (**C**). The results were obtained in three independent experiments. Error bars represent standard deviations.

**Figure 3 molecules-27-08181-f003:**
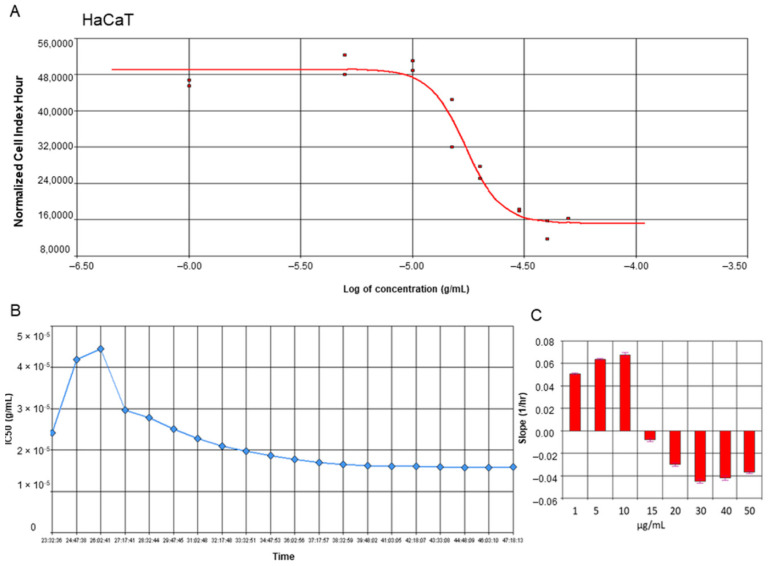
RTCA profiles of cell index (**A**), IC_50_ values (**B**), and slope of changes (**C**) in cytotoxic effect of yamogenin on the HaCaT cells. The cells were incubated with the compound for 24 h, and normalized cell index vs. concentration of yamogenin was calculated by RTCA system (**A**). The IC_50_ values of yamogenin were obtained at every time point of the experiment (every 30 min) (**B**). The slope of changes in cytotoxic effect of yamogenin was calculated for every used concentration of the compound during the experiment (**C**). The results were obtained in three independent experiments. Error bars represent standard deviations.

**Figure 4 molecules-27-08181-f004:**
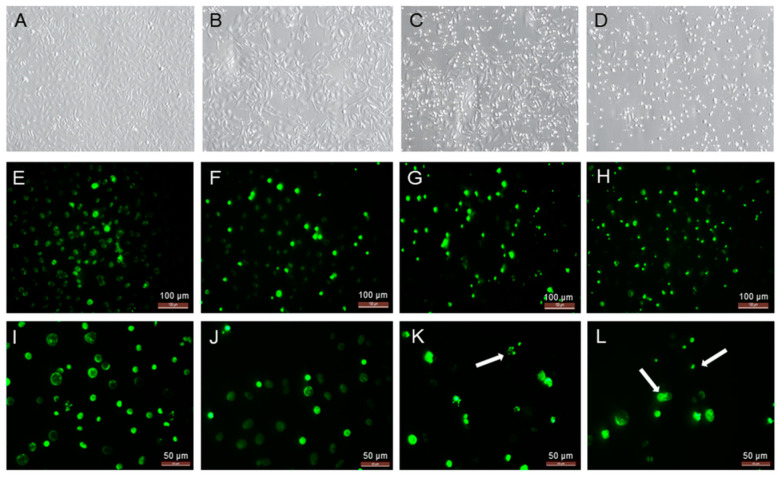
The SKOV-3 cells treated with yamogenin for 24 h and processed with Hoechst 33342 dye. The cells were incubated with ethanol (0.7% (*v*/*v*)—a control, (**A**,**E**,**I**)), yamogenin at concentrations of 20 (**B**,**F**,**J**), 50 (**C**,**G**,**K**), and 70 µg/mL (**D**,**H**,**L**), and then stained with fluorescent Hoechst 33342 dye (**E**–**L**). The cells were observed under 200× magnification (**A**–**H**) and 400× magnification (**I**–**L**). Arrows indicate the condensation of chromatin in the treated cells.

**Figure 5 molecules-27-08181-f005:**
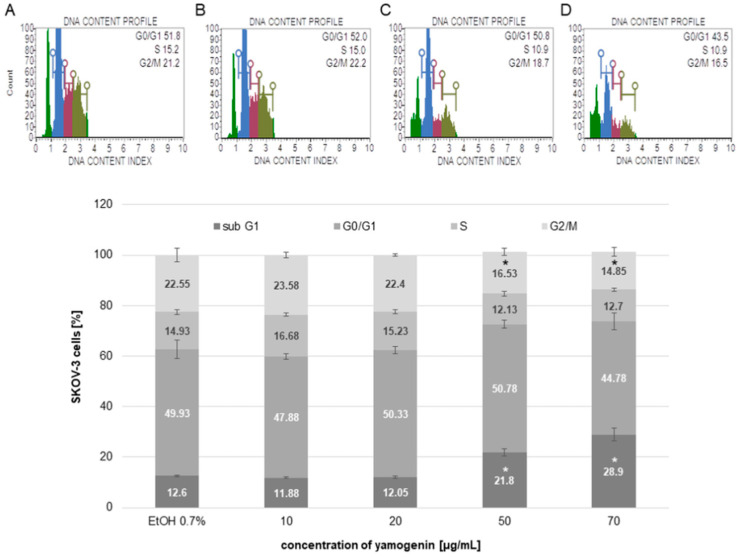
The arresting of SKOV-3 cell cycle induced by yamogenin. The ethanol and compound were added to the cells at concentrations of 0.7% ((*v*/*v*), a control, (**A**)) and 10, 20 (**B**), 50 (**C**), and 70 µg/mL (**D**), respectively. After 48 h, the cells were analyzed with flow cytometry. Green color on (**A**–**D**) represents the sub-G1, blue—the G0/G1, purple—the S, and khaki—the G2/M population of the cells. The values represent the means ± SD obtained from three independent experiments. Error bars represent standard deviations. Significant differences relative to the control are marked with an asterisk (the Student’s *t*-test, * *p* < 0.05).

**Figure 6 molecules-27-08181-f006:**
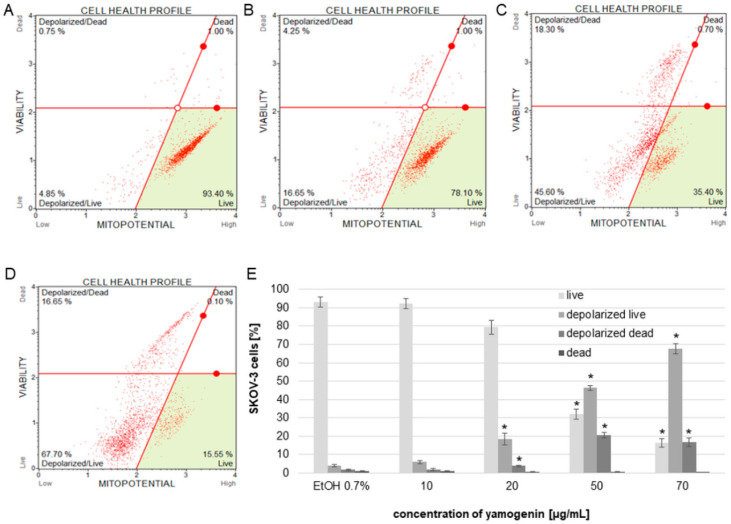
The changes in MMP of SKOV-3 treated with yamogenin for 24 h. The cells were incubated with ethanol (0.7% (*v*/*v*)—a control, (**A**)) and yamogenin concentrations of 10, 20 (**B**), 50 (**C**), and 70 µg/mL (**D**), respectively. The amount of live and dead depolarized cells was obtained with flow cytometry and is presented as percentage of the cells in the treated population (**E**). The values represent the means ± SD of three independent experiments. Error bars represent standard deviations. Significant differences relative to the control are marked with an asterisk (the Student’s *t*-test, * *p* < 0.05).

**Figure 7 molecules-27-08181-f007:**
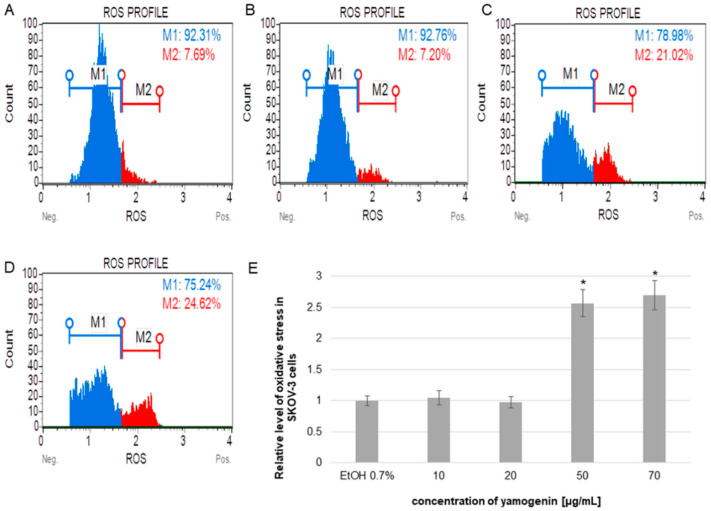
The relative level of oxidative stress production in SKOV-3 cells treated with yamogenin. The cells were incubated with ethanol (0.7%, (*v*/*v*)—a control, (**A**)), and yamogenin at concentrations of 10, 20 (**B**), 50 (**C**), and 70 µg/mL (**D**). The results were obtained with flow cytometry and are presented as the means ± SD of three independent experiments (**E**). Error bars represent standard deviations. Significant differences relative to the control are marked with an asterisk (the Student’s *t*-test, * *p* < 0.05).

**Figure 8 molecules-27-08181-f008:**
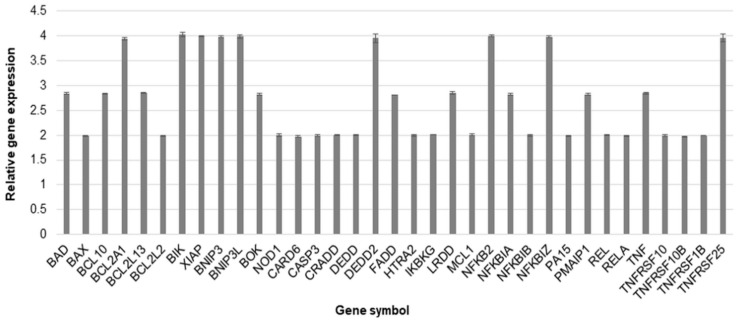
The relative gene expression at mRNA level in SKOV-3 cells treated with yamogenin. The cells were incubated with ethanol (0.4% (*v*/*v*)—a control) and the compound at a concentration of 40 µg/mL for 24 h. The results were obtained with Real-Time PCR, and values represent the means ± SD of three independent experiments. Error bars represent standard deviations. The expression of genes was normalized to HPRT1 endogenous control gene, and their levels are presented as a fold-change over the value 1.0 (control).

**Figure 9 molecules-27-08181-f009:**
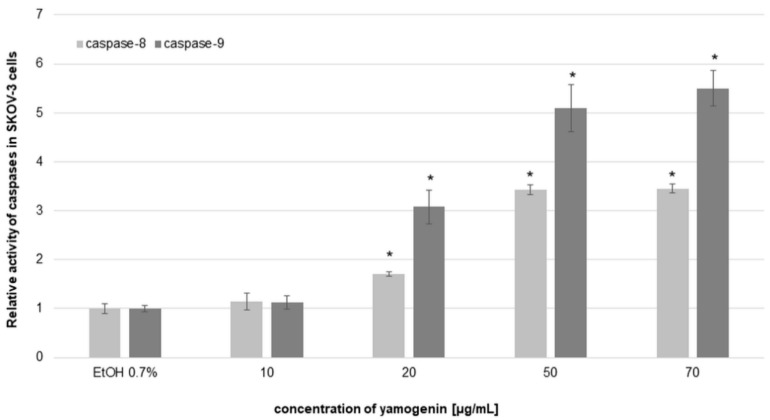
The relative activity level of caspase-8 and -9 in SKOV-3 cells treated with yamogenin. The compound and ethanol were added to the cells at concentrations of 10–70 µg/mL and 0.7% (*v*/*v*), respectively, for 24 h. The results were obtained with luminometry, and the values represent the means ± SD obtained from three independent experiments. Error bars represent standard deviations. Significant differences relative to the control are marked with an asterisk (the Student’s *t*-test, * *p* < 0.05).

**Figure 10 molecules-27-08181-f010:**
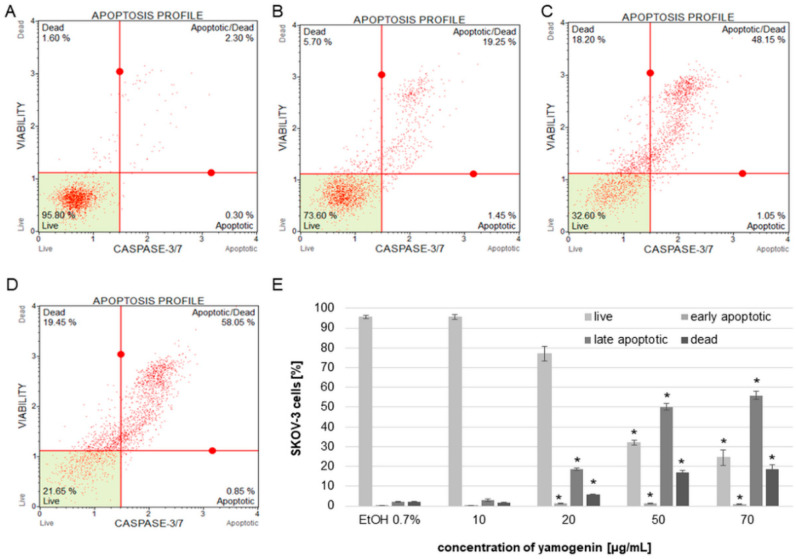
The activity of caspase-3/7 in SKOV-3 cells treated with yamogenin presented as percentage of the apoptotic cell populations. The cells were incubated with ethanol (0.7% (*v*/*v*)—a control, (**A**)) and the compound at concentrations of 10, 20 (**B**), 50 (**C**), and 70 µg/mL (**D**). The amounts of live, apoptotic, and dead cells were obtained with flow cytometry and are presented as percentage of the cells in the treated population (**E**). The values represent the means ± SD of three independent experiments. Error bars represent standard deviations. Significant differences relative to the control are marked with an asterisk (the Student’s *t*-test, * *p* < 0.05).

**Figure 11 molecules-27-08181-f011:**
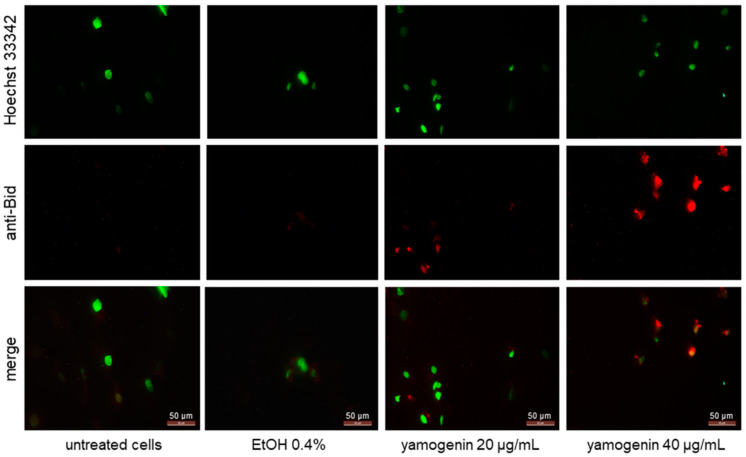
Yamogenin activates Bid in SKOV-3 cells. The cells were incubated with ethanol (0.4% (*v*/*v*)—a control) and yamogenin at concentrations of 20 and 40 µg/mL for 24 h and then stained with primary anti-Bid antibody (1:300), secondary anti-rabbit IgG Alexa Fluor 594 antibody (1:1000, red fluorescence), and Hoechst 33342 dye (green nuclei). The results were observed under a fluorescent microscope at 400× magnification.

**Figure 12 molecules-27-08181-f012:**
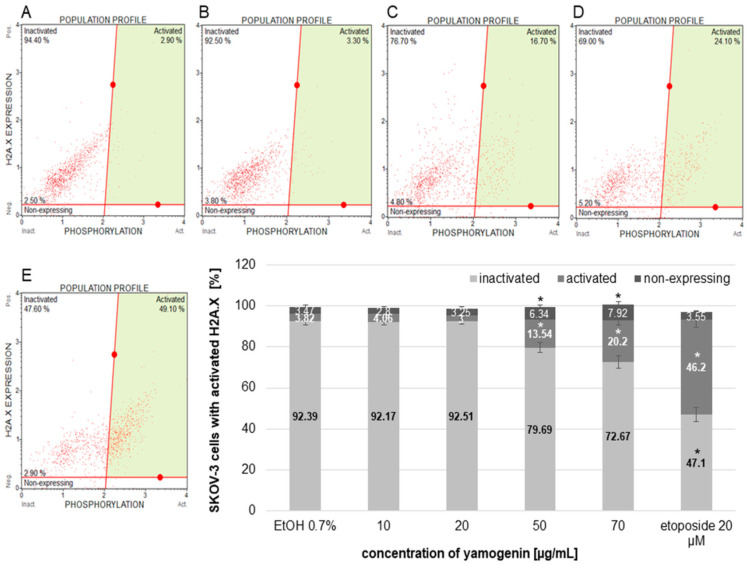
The activation of H2A.X in the SKOV-3 cells treated with yamogenin. The cells were incubated with ethanol (0.7% (*v*/*v*)—a control, (**A**)) and the compound at concentrations of 10, 20 (**B**), 50 (**C**), 70 µg/mL (**D**), and etoposide (**E**) as a positive control for 24 h. The results were obtained with flow cytometry, and values represent the means ± SD obtained from three independent experiments. Error bars represent standard deviations. Significant differences relative to the control are marked with an asterisk (the Student’s *t*-test, * *p* < 0.05).

**Figure 13 molecules-27-08181-f013:**
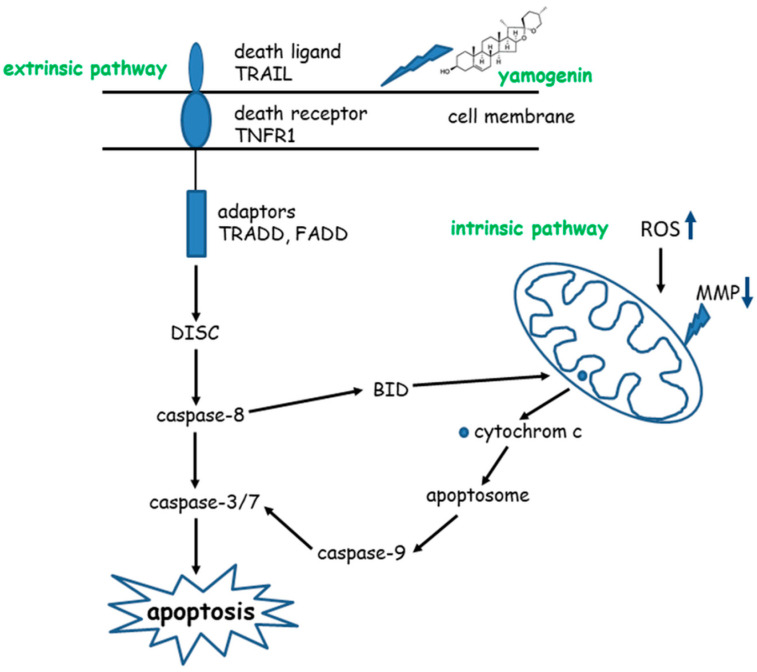
Pathways of apoptosis induced by yamogenin in SKOV-3 cells.

## Data Availability

Not applicable.

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
