# Peer review of "Yamogenin-Induced Cell Cycle Arrest, Oxidative Stress, and Apoptosis in Human Ovarian Cancer Cell Line"

_molecules, 2022, doi:10.3390/molecules27238181_

Round 1

Reviewer 1 Report

The current study clearly demonstrates that the anti-tumor effect of yamogenin in human ovarian cancer SKOV-3 cells in vitro. These results reveal that the compound triggers cell cycle arrest, overproduction of oxidative stress, mitochondrial membrane depolarization, caspase-8, -9, -3/7 and Bid activation, and DNA damage. Several issues as follows:

1: Please added the scale bars for Figure 3 and Figure 9.

2: In Figure 4 and 10, please supplement the picture of the flow cytometer results, not just the final statistical results.

3. In page 5 (2.3), please add Spaces for numbers and operation symbols such as “4.1±0.75, 5.99±0.96, 18.48±3.16, 46.28±1.44, 67.70±2.67”.

4: The panel of assays in the TaqMan® Array 96-well Human Apoptosis Plate (ThermoFisher Scientific) targets genes from both of the signaling pathways that initiate mammalian apoptosis, the death receptor regulated pathway and the BCL-2 family pathway. The authors should first perform this experiment after completing the cellular level study. This experiment can complete the screening of changes in the expression of a large number of apoptosis-related genes. Since apoptosis is closely related to protein cleavage activation, transfer, and phosphorylation modification, the next protein-related experiments are carried out for further verification.

Therefore, the experimental design of this study is a bit messy, and it is recommended to reorganize.

5: The description of mechanism of action is a bit messy, it is recommended to add the putative mechanism of action diagram.

Author Response

November 16, 2022

Dear Editor,

We would like to thank the Reviewer for critical reading this manuscript and valuable suggestions. We have carefully considered all of the suggestions and made the appropriate corrections and additions as suggested (marked in red in the revised manuscript).

Reviewer 1

The current study clearly demonstrates that the anti-tumor effect of yamogenin in human ovarian cancer SKOV-3 cells in vitro. These results reveal that the compound triggers cell cycle arrest, overproduction of oxidative stress, mitochondrial membrane depolarization, caspase-8, -9, -3/7 and Bid activation, and DNA damage. Several issues as follows:

Comment 1: Please added the scale bars for Figure 3 and Figure 9.

Response: Thank you for your valuable comments. These scale bars have been added in the figures (now Fig. 4 and 11).  

Comment 2: In Figure 4 and 10, please supplement the picture of the flow cytometer results, not just the final statistical results.

Response: The flow cytometry results have been added in figures (now Fig. 5 and Fig. 12).

Comment 3: In page 5 (2.3), please add Spaces for numbers and operation symbols such as “4.1±0.75, 5.99±0.96, 18.48±3.16, 46.28±1.44, 67.70±2.67”.

Response: This spaces have been added in the text (now page 6).  

Comment 4: The panel of assays in the TaqMan® Array 96-well Human Apoptosis Plate (ThermoFisher Scientific) targets genes from both of the signaling pathways that initiate mammalian apoptosis, the death receptor regulated pathway and the BCL-2 family pathway. The authors should first perform this experiment after completing the cellular level study. This experiment can complete the screening of changes in the expression of a large number of apoptosis-related genes. Since apoptosis is closely related to protein cleavage activation, transfer, and phosphorylation modification, the next protein-related experiments are carried out for further verification.

Therefore, the experimental design of this study is a bit messy, and it is recommended to reorganize.

Response: Thank you for this comment. We have reorganized the manuscript and data showing genes expression are now before analyses of activation of caspase-3/7/8/9, Bid, and H2A.X.   

Comment 5: The description of mechanism of action is a bit messy, it is recommended to add the putative mechanism of action diagram.

Response: The proposed mechanism of action of yamogenin has been added in the Discussion section (Figure 13, page 13).

We thank once again the Reviewer for all valuable suggestions and hope that the revised manuscript is now appropriate for publication in Molecules.

Sincerely,

Justyna Stefanowicz-Hajduk, Ph.D., assistant professor,

Department of Biology and Pharmaceutical Botany,

Medical University of Gdańsk, Gdańsk, Poland

Reviewer 2 Report

Moleculars 

Manuscript ID: molecules-2026621

Title: Yamogenin-induced cell cycle arrest, oxidative stress and apoptosis in human ovarian cancer cell line

Dear Authors: 

Congratulations on writing such an interesting effect of novel saponin, Yamogenin. This manuscript evaluates the anti-cancer activity of Yamogenin, a stereoisomer of the plant-derived saponin Diosgenin, and tried to clarify the mechanism of anti-tumor effects. To this end, authors were using in vitro experimental system with SKOV-3 cells, a cultured line of human ovarian cancer cells. Their results show that Yamogenin induces apoptosis in SKOV-3 cells, which occurs through an intracellular pathway by mitochondrial membrane, ROS stress, and caspase activation, and an external pathway that upregulates sensitivity to extracellular stimulation such as TNF family and death signaling. These results are very interesting and well summarized, as it is a combined action of pathways common to previously known other saponins and Diosgenin.

However, there are several problems that listed below.

The major weak point of this manuscript is the lack of data on a clear comparison with Diosgenin and other saponins. It is not clear what is advantageous or clearly distinctive about the anti-tumor activity of Yamogenin compared to previously known saponins. I think that the pathological significance of clarifying the anticancer activity of Yamogenin should be clearly demonstrate in the main text.

The following are my comments and suggestions:

1. The conformational structures of Yamogenin and Diosgenin are needed in first figure.

2. It would be better to have data on the comparison of the effect of Diosgenin and Yamogenin in inhibiting cell proliferation and the difference in the mechanism of action. At the very least, I think that a clear comparison between Diosgenin and Yamogenin is necessary in the discussion part.

3. To discuss extracellular stimulation, the effects of Yamogenin on other organs should also be discussed; consider whether Yamogenin has the ability to cause the release of TNF family and FAS ligands from other tissues of the immune system and or cells around cancer.

4. Figure 11 shows increased gene expression after treatment with Yamogenin, but were there any factors that were conversely suppressed, or were antioxidant enzymes such as SOD, catalase, or phosphatases affected ?

Author Response

November 16, 2022

Dear Editor,

We would like to thank the Reviewer for critical reading this manuscript and valuable suggestions. We have carefully considered all of the suggestions and made the appropriate corrections and additions as suggested (marked in red in the revised manuscript).

REVIEWER 2

Comment 1: The major weak point of this manuscript is the lack of data on a clear comparison with Diosgenin and other saponins. It is not clear what is advantageous or clearly distinctive about the anti-tumor activity of Yamogenin compared to previously known saponins. I think that the pathological significance of clarifying the anticancer activity of Yamogenin should be clearly demonstrate in the main text.

Response: Thank you for these comments. The significance of clarifying the anticancer activity of yamogenin as well as better comparison of the action of yamogenin with diosgenin and other saponins we have made in the manuscript (Introduction and Discussion section).    

Comment 2: The conformational structures of Yamogenin and Diosgenin are needed in first figure.

Response: We have added these structures in the manuscript (Figure 1, page 2).

Comment 3: It would be better to have data on the comparison of the effect of Diosgenin and Yamogenin in inhibiting cell proliferation and the difference in the mechanism of action. At the very least, I think that a clear comparison between Diosgenin and Yamogenin is necessary in the discussion part.

Response: This comparison has been made in the Discussion section (page 13 and 14).

Comment 4: To discuss extracellular stimulation, the effects of Yamogenin on other organs should also be discussed; consider whether Yamogenin has the ability to cause the release of TNF family and FAS ligands from other tissues of the immune system and or cells around cancer.

Response: The study on yamogenin, generally, is very limited. Our team started anticancer tests in vitro with this compound due to very little knowledge about the cytotoxic activity of this saponin. Up to now, we have not found information about potential action of yamogenin on organs/cells in vivo or ex vivo. Also, we have not found papers describing the effect of yamogenin in tissues in the case of affecting TNF, FAS pathway. The future study is needed in this field due to the fact that steroidal saponins may act as a strong antioxidant, anti-inflammatory and angiogenic factors in different tissues.

Comment 5: Figure 11 shows increased gene expression after treatment with Yamogenin, but were there any factors that were conversely suppressed, or were antioxidant enzymes such as SOD, catalase, or phosphatases affected?

Response: In this study, we observed and described that the expression of genes such as FAS, TNFRSF21, and TNFSF10 were significantly lower than the control (at least 0.5 times lower, page 7). For other genes from the tested TaqMan Human Apoptosis panel (Table S1, Supplementary Materials) we did not observe changes in the expression at mRNA level after treating the cells with yamogenin. The TaqMan panel of 96 genes, used in this study, does not contain assays for the analysis of expression of genes coding catalases, phosphatases etc. However, we are planning in the near future other experiments on action of yamogenin on other cellular factors and their activation/suppression and also on activity of enzymes in vitro, including antioxidant activity tests.

We thank once again the Reviewer for all valuable suggestions and hope that the revised manuscript is now appropriate for publication in Molecules.

Sincerely,

Justyna Stefanowicz-Hajduk, Ph.D., assistant professor,

Department of Biology and Pharmaceutical Botany,

Medical University of Gdańsk, Gdańsk, Poland

Round 2

Reviewer 2 Report

Comments to authors:

After a careful review of this resubmitted manuscript, all my reviewer`s comments of the weakness in previous version are clearly improved in this version.  So, I recommended this review to be published in the Molecules.